



# On the effects of inter-farm interactions at the offshore wind farm Alpha Ventus

Vasilis Pettas[1], Matthias Kretschmer[1], Andrew Clifton[1], and Po Wen Cheng[1]

[1]Stuttgart Wind Energy (SWE), University of Stuttgart, Allmandring 5b, 70569 Stuttgart, Germany

**Correspondence:** Vasilis Pettas (pettas@ifb.uni-stuttgart.de)

**Abstract.** The energy transition means that more and more wind farms are being built in favorable offshore sites like the North Sea. The wind farms affect each other as they interact with the boundary layer flow. This phenomenon is a topic of current research by the industry and academia as it can have significant technical and financial impacts. In this study we use data from the Alpha Ventus wind farm site to investigate the effects of inter-farm interactions. Alpha Ventus is the first offshore German wind farm located at the North Sea with a fully equipped measurement platform FINO1 in the near vicinity. We look at the effects on the wind conditions measured at FINO1 before and after the beginning of operation of the neighboring farms. We show how measured quantities like turbulence intensity, wind speed distributions, and wind shear are evolving from the period where the park was operating alone in the area to the period where farms were built and operate in close proximity (1.4-15 km). Moreover, we show how the wind turbine performance is affected using data from a turbine that is in the vicinity of the mast. The results show the wake effects in the directions influenced by the wind farms according to their distance with increased turbulence intensity, reduced wind speeds, and increased structural loading.

## 1 Introduction

The reduction of produced emissions and the transition to renewable energy sources require a large increase in the installed capacity of wind. This leads to more and larger wind farms being built with ever-increasing turbine sizes. Offshore sites have a lot of benefits compared to onshore -i.e. higher wind speeds with lower turbulence, higher social acceptance, larger space availability, and larger energy density due to the possibility to install larger machines. Thus, offshore sites with favorable wind and soil/depth conditions, like the North Sea, are being populated with wind farms that have to be spaced as close as possible.

Similar to single wind turbine wakes, the wind farms as a whole interact with the atmospheric boundary layer and create wakes that are propagated downstream (Porté-Agel et al., 2020). This phenomenon is more prominent in offshore farms where the machines used are larger, the ambient turbulence intensity is lower, and the surface roughness is lower than onshore sites. These effects need to be modelled and considered when planning the siting of wind farms as they can have a large impact on





the operating conditions experienced by the neighboring wind farms. Neglecting such effects can lead to large deviations on the annual energy production (AEP) estimates as well as the life time of the structural components (Lundquist et al., 2019).

Therefore, understanding these inter-farm interactions has been a topic of interest for research as well as the industry. Previous studies based on airborne measurements (Cañadillas et al., 2020; Platis et al., 2018), have identified wind speed deficits and increases in turbulence downstream of offshore wind farms and clusters. They showed that these effects are visible up to a level of 50 km downstream, especially in neutral atmospheric conditions. In Ahsbahs et al. (2020), measurements with synthetic aperture radars (SAR) and Doppler radars were performed showing a speed deficit of 4% to 8% up to 10 km
downstream of the farms. The work of Christiansen and Hasager (2005) used also SAR measurements to investigate wind farm wakes at the North Sea and the Baltic sea. They reported a speed deficit in the near wake, with full recovery at distances of 5-20 km depending on the free stream wind speed, the atmospheric stability, and the number of operating turbines. In the study of Mittelmeier et al. (2017), turbine data were used to identify the magnitude of the wake effects. They reported a detectable wind speed deficit and wake-induced turbulence due to neighboring farms at a distance of 4.2 km (38.8 D).

Moreover, numerical studies have been carried out to understand the underlying mechanisms of wind farm wakes. In Wu and Porté-Agel (2017) LES simulations were performed to correlate atmospheric stratification, farm size, and layout with the flow inside and around a wind farm. They identified velocity deficits at a level of 3.5% compared to free stream values at a 10 km distance for small vertical temperature gradients (1 K/km) with the wake-induced turbulence propagating up to 10 km. Moreover, for a higher gradient of 5 K/km, they report a full wake recovery at a distance of 5km. Lu and Porté-Agel (2015)
studied the effects of large wind farms on the atmospheric boundary layer, showing that it is affected due to the enhanced mixing caused by the wakes.

The need to include these effects in practical engineering studies has led to the development of engineering wind farm wake models with varying fidelity and requirements. Emeis (2009) suggested a simple analytical model to calculate the speed deficit accounting for atmospheric stability, surface roughness, the turbine's thrust coefficient, and the Monin-Obukhov length. A
calibration and evaluation of the model can be found in Platis et al. (2020). In Nygaard et al. (2020) an engineering model accounting for both speed deficits and wake-induced turbulence is suggested. It is based on the Jensen-Katic model (Katic et al., 1987), extended to include wake-induced turbulence and validated against measurements for power production.

In this context, the goal of the present work is to investigate the effects of inter-farm wakes by analyzing measurement data. More specifically, metocean data from the FINO1 measurement platform along with SCADA and loads data from the closest
machine of Alpha Ventus (AV) to the mast are analyzed for the period 2010-2019. Until 2015, AV was the only wind farm operating in the area. This setup allows us to observe how the metocean conditions - as measured by FINO1 - have changed in relation to the surrounding wind farm construction and how these changes impact the turbine's response at AV. The data are analyzed per direction and wind speed, on an annual basis. This will give insight to researchers working on modelling the inter-farm interactions as well as to practitioners focusing on optimizing wind farm siting and planning. Furthermore, the data
presented here can be used by researchers doing research related to AV in order to represent the metocean conditions during the different operational periods of the project.





The rest of the paper is structured as follows: in section 2 we describe the site and the locations of the farms and the met mast around the Alpha Ventus site. In section 3 the measurement equipment and the data processing methods are discussed. Section 4 presents the results in terms of metocean conditions and turbine responses followed by a discussion on the findings in section 5.

## 2 Site description

Alpha Ventus (AVh, 2020) is the first German offshore farm commissioned in 2010 and is located in the North Sea close to the island of Borkum. It consists of 12 fixed bottom wind turbines with a rated power of 5 MW. Half of the turbines are manufactured by REpower (renamed to Senvion) having a jacket support structure and the rest are manufactured by Adwen and use a tripod substructure. The research projects at AV are supported by the initiative Research at Alpha Ventus (RAVE) (RAV, 2020) by coordinating the research activities and providing measurement data.

FINO1 (FIN, 2020) is a research measurement platform including a fully equipped met mast, erected in 2004 at the North Sea. It is located close to AV, at a 400 m (3.2D) distance from the AV4 turbine. The data from both FINO1 and AV are processed and made available in one database (BSH, 2020) operated by the Federal Maritime and Hydrographic Agency (BSH). Data have been collected since the pre-construction phase (2004-2010) of AV and also during the operational phase from 2010 to today.

As it was initially planned, new wind farms have been built around AV over time, operating in different distances and directions. Today, AV can be seen as a part of a larger wind farm cluster consisting of 5 wind farms equipped with similarly sized turbines. This allows us to observe how the site conditions have evolved during the years due to the inter-annual variability but also due to the presence of the neighboring wind farms.

An overview with all the wind farm information, the distances, and the directions relative to FINO1 are given in table 1. A sketch intended to give the reader an impression of the topology of the area is given in figure 1. In 2015, two new wind farms started operating in the area. The closer one, Borkum Riffgrund 1 (BR1) (Rif, 2020) is located south-west of AV in distances between 2.8 and 7.6 km from FINO1 and the relevant directions are 70 to 244 deg (considering FINO1 as the origin of a clockwise system with north pointing up at 0 deg). BR1 consists of 78 turbines of 4 MW with a rotor diameter of 120 m. The second, Trianel Borkum 1 (TB1) (Tri, 2020) is located north-east of AV in distances between 6 and 14 km and directions 253-315 deg relatively to FINO1. The total installed capacity is 200 MW consisting of 40 turbines with rotor diameters of 116 m.

At the beginning of 2019, the Merkur (MRK) (Mer, 2020) wind farm was commissioned. Merkur is the closest farm to FINO1 with relative distances between 1.4 and 8 km with a relevant azimuth sector of 235-45 deg. MRK consists of 66 turbines rated at 6 MW with 150 m rotor diameters and a total capacity of 396 MW. The second part of the Borkum Riffgrund project, the Borkum Riffgrund 2 (BR2) wind farm started operating mid-2019. It is the largest wind farm in the area with 448 MW capacity consisting of 8 MW machines with 164 m rotor diameters. The relevant directions to FINO1 are 80-250 degrees



in distances between 9 and 13 km. Finally, in 2020 the extension of the Trianel Borkum project, Trianel Borkum 2 wind farm
started operating but it is not in the scope of this study as we use data up to 2019.

Most of the information (e.g. construction/commissioning dates, geographic locations) shown here are gathered from publicly available sources which could not be fully verified. As it is also explained in the following section regarding the data, some uncertainties exist due to imprecise information in dates, coordinates, maintenance logs, etc., which led us to be more conservative on the data filtering and consider them in the interpretation of the results.

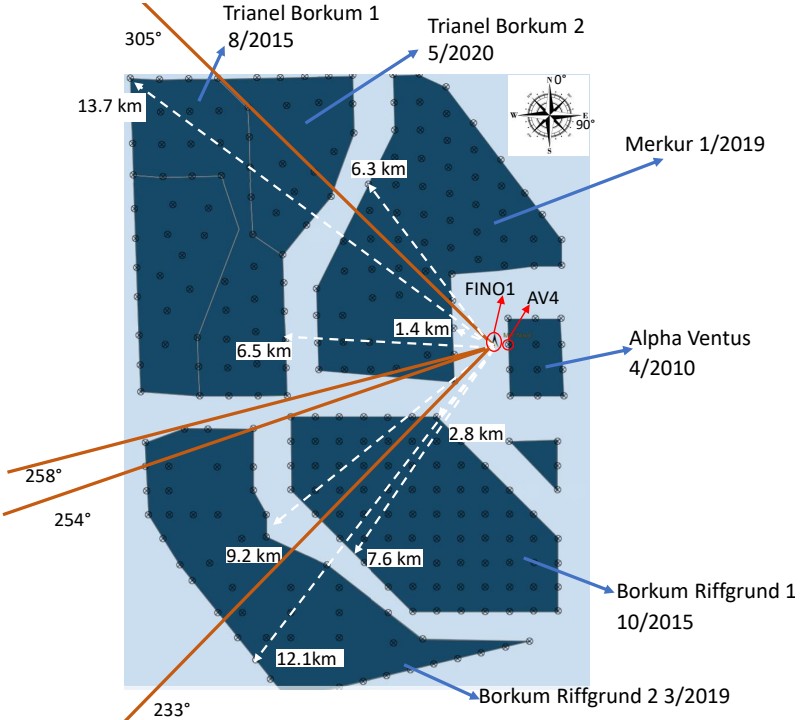

**Figure 1.** Layout of Alpha Ventus, FINO1 and the surrounding wind farms including relative directions and distances. Origin is the FINO1 platform. Background layer used with permission from 4COffshore

## 3   Measurement equipment and data processing

In this section we present the measurement equipment used, the data filtering/calibration approaches used, and the post-processing methods applied to obtain the results shown in the next section. All data were obtained through the RAVE database (BSH, 2020) which is publicly available under user agreements. The data used in this study were from the period 01.01.2011-31.12.2019.

The metocean data were collected from the FINO1 platform. The data were provided as 10-minute statistics with corrections and calibration values applied as instructed by the providers. As these data are commonly used in the research community,



**Table 1.** Overview of the wind farms at the site and the relevant distances and azimuth directions to FINO1.

| Wind Farm Name | WT nominal power [MW] | Rotor diam. [m] | Total WT | OEM | Operation begins | Min. distance to FINO1 [km] | Azimuth sector [deg] |
|---|---|---|---|---|---|---|---|
| Alpha Ventus [AV] | 5 | 126/116 | 12 | Senvion/Adwen | 4/2010 | 0.4 | 30-170 |
| Merkur [MRK] | 6 | 150 | 66 | GE | 1/2019 | 1.4 | 230-40 |
| Trianel Borkum 1 [TB1] | 5 | 116 | 40 | Adwen | 8/2015 | 6.5 | 255-305 |
| Trianel Borkum 2 [TB2] | 6.25 | 152 | 32 | Senvion | 5/2020 | 6.7 | 260-325 |
| Borkum Riffgrund 1 [BR1] | 4 | 120 | 78 | Siemens | 10/2015 | 2.8 | 155-255 |
| Borkum Riffgrund 2 [BR2] | 8 | 164 | 56 | Vestas | 3/2019 | 9.2 | 170-260 |

we do not discuss all the sensors available and the specifics of the setups here. A thorough overview of the database and data quality considerations can be found in Quaeghebeur and Zaaijer (2020). In the study of Hübler et al. (2017), a comprehensive database of aggregated fitted distributions for different quantities is presented for the period 2004-2016. These data have been

also used for examining the influence of stability on the loading spectrum of the Alpha Ventus turbines (Kretschmer et al., 2019) as well as for validation of aeroelastic farm-wide simulation tools (Kretschmer et al., 2021).

For the wind conditions, we used the cup anemometers mounted at heights from 30 to 100 m above sea level (a.s.l.) at increments of 10 m. The wind speed and turbulence intensity (TI) measurements were taken from the top anemometer at 101.5 m. This is higher than the 92 m hub height of the Alpha Ventus turbines but was selected since there are fewer mast

shadowing effects in the directions of interest (more information on this can be found in Westerhellweg et al., 2011). The data were corrected for shadow effects and calibration according to communication with the data providers and as explained in Westerhellweg et al. (2011). The wind shear was calculated from all available heights assuming a power law exponent. The shear exponent $\alpha$ was fitted to the data by minimizing the least squares difference with the measured values. The wind direction statistics were obtained by the wind vane at 90 m with the relevant corrections applied as discussed in Westerhellweg et al.

(2010). The temperature and pressure data were also obtained as 10-minute statistics. We used the thermometers on the mast at 34 and 52 m a.s.l. and the pressure was obtained by the barometer at 21 m a.s.l. The oceanographic data used here were also 10-minute statistics and included sea surface temperature from the buoy and significant wave height ($H_s$) and peak period ($T_p$) from the radar mounted at the platform.

The turbine data used come from the AV4 turbine which is manufactured by Senvion and has a rated power of 5 MW, a hub

height of 92 m, and a rotor diameter of 126 m. It is located at 92 degrees azimuth relative to FINO1 at a distance of 3.2 D. The data were provided at a 50 Hz sampling rate. For the tower base loads, we used the strain gauges located above the transition piece and combined them with the nacelle yaw position signals to derive the fore-aft loads. Moreover, from the SCADA signals, we obtained the nacelle yaw position, blade pitch angle, generator power, and generator speed. The calibration of the load and nacelle yaw sensors was done according to the provided values as well as using events where the turbine was not operating



and the nacelle was rotated 360 degrees. Finally, small corrections on the calibrations factors were done to compensate for the sensor drift observed over time.

In order to filter the metocean statistics data we applied multiple criteria. The quality flags provided by the database were used. Data blocks with availability less than 90% for the 10 min period were rejected. Appropriate thresholds for minimum, maximum, and standard deviation values were implemented as an additional filtering criterion. For the high-resolution turbine

data, we applied the same filters for availability and statistics as before. Additionally, we only accepted events where the nacelle yaw position and the met mast wind direction had a difference smaller than 3 degrees to avoid including events where the turbine was misaligned with the mast. Similarly, we rejected events where the nacelle position signal's standard deviation and the difference between maximum and minimum value were higher than specific thresholds to avoid including the effect of yawing in the calculated loads. Furthermore, there were many periods where the turbine was operated with curtailed power

which affects loads (see e.g. Kretschmer et al. (2018)). We filtered out these periods by applying a filter involving combined thresholds for pitch angle, generator speed and power, and wind speed to make sure that the machine is operated invariably in full power. Turbine data from 2019 were not available to us at the time of this study.

For the post-processing of the loads, we used the rainflow algorithm and Miner's rule with a Wöhler exponent of 4 and a reference cycle number of 600 to derive the 1 Hz Damage Equivalent Loads (DEL). Due to the non-disclosure agreement

with the turbine manufacturer, the DEL values shown are normalized with values close to the maximum. For the wind speed distributions, we binned the data and fitted the Weibull parameters k and C using least squares. In all the plots that include bands, they represent the $15^{th}$ and $85^{th}$ percentiles. Finally, we don't show the scatter but only the mean value and the band to avoid congestion and improve clarity for the reader.

Regarding the periods of measurements, we decided to use full years for all the quantities. This was done to facilitate

the analysis of the wake effects which is the main objective of this study and also to avoid bias due to seasonal variability. To achieve this, we rejected measurements that had large continuous gaps in the year and also years with less than 80% availability in total. In other analyses that are based on binning (e.g. directional or wind speed bins) we accepted only bins that included 25 points or more to avoid statistical bias.

There are some known sources of uncertainty affecting the results shown here. We don't have access to the service log of the

turbines, hence we are not aware of maintenance or replacement activities on the turbine components (or measurement devices) that might have influenced the results. We tried to compensate for these with the filtering approaches mentioned earlier. We also don't have access to operational information of the neighboring farms in order to know when the farms might have stopped operating or when they were curtailed which is expected to have an impact on the wake effects. Finally, availability varies significantly from sensor to sensor. Especially in turbine data where the filters required a combination of sensors, the resulting

availability was much lower.

Another known uncertainty comes from the measurements relevant to directions of 30-190 degrees. There, FINO1 is in the direct wake of Alpha Ventus. Thus, we shouldn't consider them as free-stream directions and take into account that probably wind speeds are underestimated and TI is overestimated compared to the real free-stream wind speeds seen by Alpha Ventus in this sector. Nevertheless, this sector is useful for our analysis as there have been no changes in the surroundings since AV





started operating. Thus, this sector can be used as a validation sector influenced only by the inter-annual variability and the measurement uncertainties but not by wake-related effects.

## 4 Results

The results from FINO1 and the AV4 turbine will be presented in two sections. The first is dedicated to metocean conditions and the second to the turbine's response. We are going to analyze these in terms of azimuth directions, as shown in figure 2.

This figure can be used as a reference for the reader for the positioning of the farms and the azimuth sectors relative to FINO1, which is used as the origin. Additionally, we use a color code with shades of blue for the years where Alpha Ventus was the only operating farm in the area (2011-2014) and shades of red for the years 2015-2018 where TB1 and BR1 farms were also operating. The yellow color refers to the year 2019 when MRK and BR2 wind farms started operating too.

We will focus the analysis on the sector 200-320 degrees as its the relevant one for investigating the inter-farm wakes. In the

directions 30-170 degrees the mast measurements are directly influenced by the wakes of AV. The directions around 180 and 360 degrees are influenced by the mast's own shadowing as previous studies have also shown (see e.g., Westerhellweg et al., 2010). Hence, we decided to exclude these sectors which also have a low probability of occurrence.

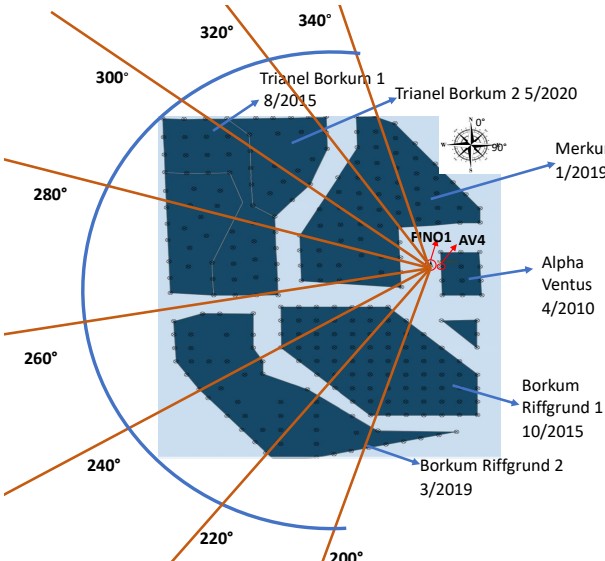

**Figure 2.** Azimuth sectors of focus and wind farms orientation layout. Background layer used with permission from 4C Offshore

### 4.1 Metocean conditions

Initially, an overview of the probability of occurrence of wind directions and wind speeds at the site can be seen by looking

at the cumulative wind rose including all periods in figure 3. The dominant directions are in the sector 200-330 degrees with





more than 58% probability, while the highest wind speeds are observed at the south west sector. This principal sector is the one potentially affected by the neighboring farms and we are going to focus our analysis on it. The measurements on the eastern directions (30-170 deg) are heavily influenced by the wake of Alpha Ventus itself and are expected to underestimate the wind speed magnitude. The probability of occurrence for this sector is about 25%.

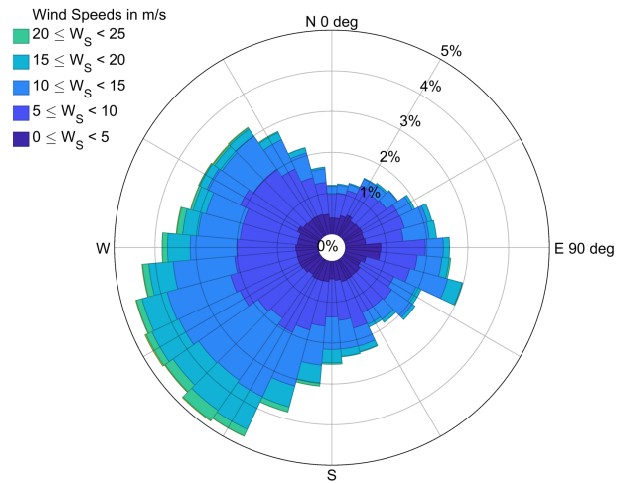

**Figure 3.** Wind rose from FINO1 cup anemometers and wind vane at 90 m. All periods are considered

We analysed the annual wind speed magnitude probability by looking at the fitted Weibull distributions. In figure 4 we show the annual distributions, considering all the wind directions, along with the relevant IEC standard (IEC, 2019) design class of the AV4 turbine (class II). The years 2015 and 2018 were not used as there were significant data gaps that could cause seasonal bias. Also, 2015 was the year that TB1 and BR1 started operating leading to a change in the measured conditions during the year. The differences between the years 2011-2014 are attributed to the inter-annual variability of the weather conditions. In

years 2016 and 2017, we observe a shift of the distributions towards lower wind speeds due to the operation of TB1 and BR1. In 2019 there is a further reduction in the wind speeds due to the operation of MRK and BR2. The expected deviations due to inter-annual variability (Pryor et al., 2018) are smaller compared to the deviations due to the wakes and a pattern of shifting towards lower speeds is observed. The IEC class II distribution suggests higher wind speeds compared to all measured periods. As mentioned earlier, all the measured distributions are expected to be slightly underestimated due to the sector influenced by

the wake of AV (30-170 degrees). This is not expected to influence the relative differences over the years since it is a constant offset.

    In table 2 we show the fitted Weibull coefficients and the calculated theoretical AEP for a single turbine operating in these conditions. The theoretical AEP is simply derived by multiplying the theoretical power curve of the NREL 5MW reference wind turbine (Jonkman et al., 2009), with the fitted Weibull distribution. It is not derived by power measurements and it is only

used to give an idea of what the reductions in wind speed would mean in terms of power production.



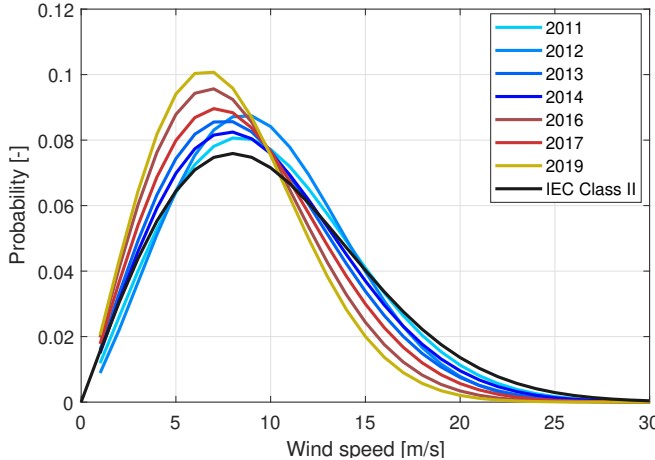

**Figure 4.** Fitted Weibull distributions of wind speed magnitudes per year including all directions.

Looking at the distribution's scale parameter C, correlated to the characteristic wind speed, we see the reduction in years 2016-2019 compared to the earlier period. The mean value is reduced by 15% between the two periods. The reduction is much larger than what is seen for the inter-annual variability (about 5%) and can be attributed to the inter-farm wake effects. As a consequence, the mean theoretical AEP is reduced by 19% with the reduction being higher in 2019 when MRK and BR2 200 were also operating reaching a level of 25%. The standard deviation of the AEP is also slightly increased in the later period. The class II expected AEP is also shown as a reference for an expectation from the site, although pre-construction site specific values would be more realistic. These results show that there can be a significant financial impact due to wind speed reductions from farm wakes and they should not be neglected.

**Table 2.** Weibull distribution coefficients and theoretical AEP calculations for the years measured. All directions are included.

| Year/Class | Weibull k parameter | Weibull C parameter | Theoretical AEP [GWh] | Mean AEP [GWh] | STD AEP [GWh] | Operating Farm |
|---|---|---|---|---|---|---|
| 2011 | 2.06 | 10.75 | 19.6 | | | AV |
| 2012 | 2.20 | 10.47 | 19.4 | 18.7 | 0.92 | AV |
| 2013 | 2.05 | 9.93 | 17.6 | | | AV |
| 2014 | 2.01 | 10.36 | 18.5 | | | AV |
| 2016 | 2.07 | 8.89 | 15.1 | | | AV,TB1,BR1 |
| 2017 | 2.02 | 9.52 | 16.5 | 15.2 | 1.25 | AV,TB1,BR1 |
| 2019 | 2.10 | 8.46 | 14.0 | | | AV,TB1,BR1,BR2,MRK |
| IEC class II | 2.00 | 11.30 | 19.5 | - | - | - |





To investigate directionally the effects of the neighboring farms on the annual wind speed distributions, we fitted the Weibull
distributions sector-wise and the results are shown in figure 5. The sector of 200-220 degrees shows the influence of BR1
leading to a high decrease in wind speeds. The distribution has not changed significantly in 2019 compared to the previous two
years. This shows that the operation of BR2, directly upstream of BR1, does not contribute to further wind speed reductions
to the micro-climate of AV. This could be a result of the sparser layout of the farm but cannot be verified without more
relevant measurements or simulations. In the sector 220-240, we see similar trends with a less pronounced reduction. In 2019,
the wind speeds are slightly reduced further, which can be attributed to the small part of MRK operating in this sector. In
sectors 260-320, BR1 and BR2 don't affect the wind distributions as they match with the earlier period. In the sector 240-
260, which is influenced by the part of BR1 that is located in distances 3.5-8 km from FINO1, we see that the wind speeds
are not affected by looking at years 2011-2017. TB1 seems not to influence the wind speeds as the sector 260-320 shows no
significant deviations in the years 2011-2017. The influence of MRK, the closest farm, is seen in the sector 240-320 degrees
with significant reductions in the year 2019.

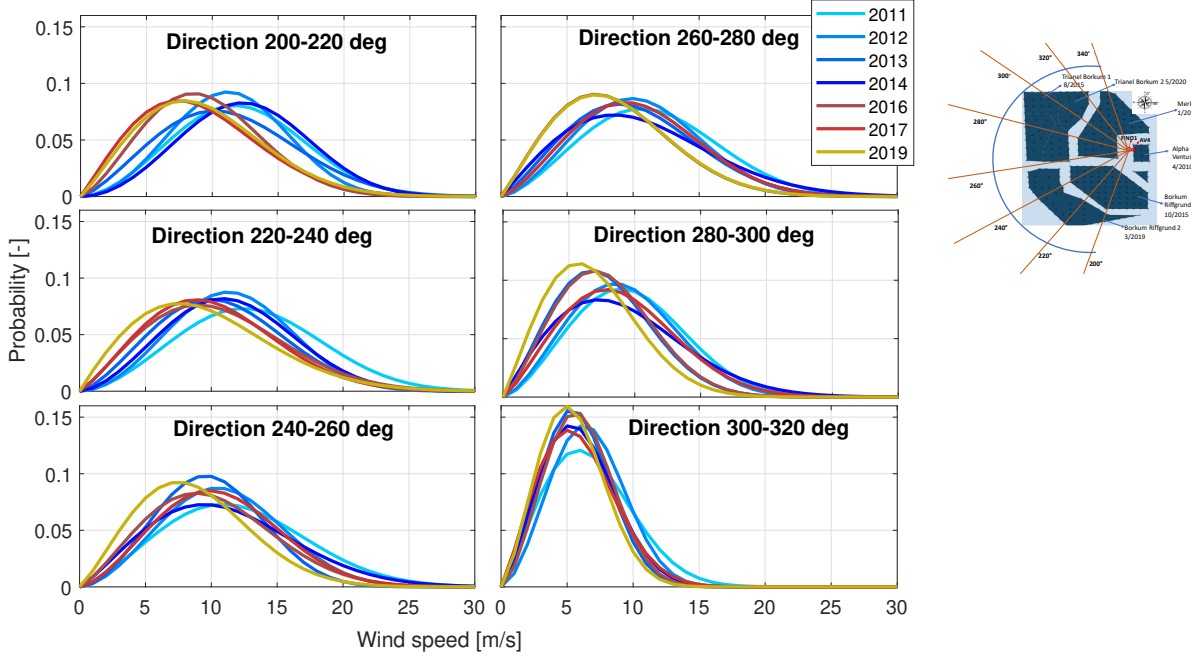

**Figure 5.** Fitted wind speed Weibull distributions per year and azimuth direction

Next, we investigate how the TI perceived by FINO1 is influenced by the wake interactions. In figure 6 we show the mean
TI per wind speed bin for the different years and azimuth directions. At the sector 200-220 degrees, the increase in TI due to
the presence of BR1 is observed. In wind speeds 6-14 m/s, an increase of 40% to 60% is seen. These results are constant in
all years for both periods suggesting that what we see is indeed attributed to wind farm wakes. The operation of BR2 in 2019
directly upstream of BR1, does not seem to lead to a further increase in TI at FINO1.





The sector 220-240 degrees shows similar trends and levels of increase in TI, with the effect of MRK in 2019 being visible as a further increase. Looking at the sector of 240-260 degrees for the years 2016 and 2017, the increase in TI due to BR1 reaches a level of 20-35%. This is smaller than the previous sector as a smaller part of the upstream farm in larger distance influences this sector. In 2019, a significant increase in TI is observed in wind speeds 4-12 m/s to a level of 70-120% compared
to the period 2011-2014. In the sectors 260-320 degrees, for the period 2016-2017 we see that the effect of TB1 is small with an increase of 2-5% in TI. In the same sectors looking at 2019, we observe a significant increase in TI due to the MRK farm. The TI is increased by 50% to 100% in wind speeds below 12 m/s. This shows that the proximity of MRK to AV (distances 1.4-7 km) has a very high impact on the wake-induced TI.

    In all cases the wake-induced turbulence is wind speed dependent. The TI increase is higher in lower wind speeds since the
upstream turbines are operating with a higher thrust coefficient (below rated operation) and the wakes dissipate faster in higher wind speeds.

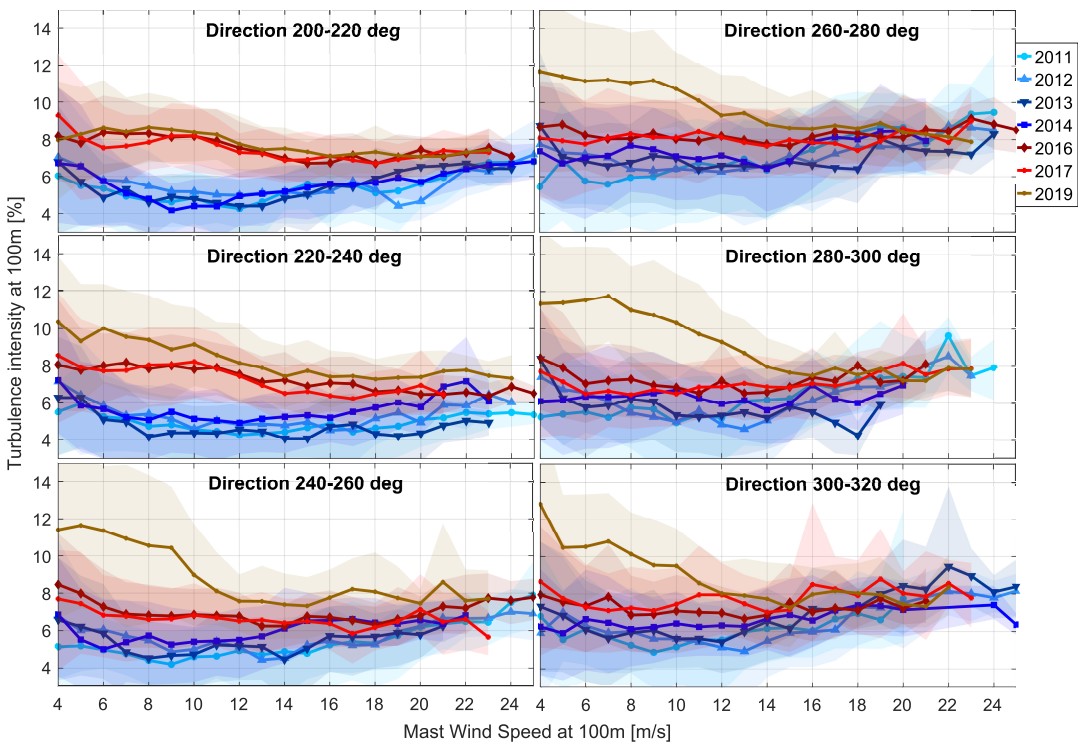

**Figure 6.** Measured turbulence intensity per wind speed for the different directions and years. Solid lines and markers represent the mean values. The band represents the $15^{th}$ and $85^{th}$ percentiles

    In figure 7 we show the mean TI for all azimuth directions (binned in 10 degrees intervals) for two wind speed bins, 6-8 m/s and 13-15m/s, representing the below and above rated operating regions. The sector 30-170 degrees is affected by the AV wakes and cannot be used to derive meaningful conclusions but shows the consistency of the measurements over the different
periods, confirming that what we see in the other sectors is attributed to inter-farm wakes. The peak observed around 270





degrees in all measurements is attributed to the shadow of the blizzard cage structure. This also explains the more increased TI levels for all years in the sector 260-280 degrees (seen also in figure 6).

In the lower wind speed bin, we notice the directional influence of the wind farms with the effect of MRK in 2019 being the most dominant. Its influence is observed already in the 220 degree direction. At 240 degrees we notice a drop where the 2019
values match the 2016-2017 values which could be explained by the sparse placing of the turbines of MRK in this direction. In the higher wind speed bin, we see a reduced effect on TI in all directions. The observed level of increase in TI is almost the same in the years 2017-2019. This suggests that even in close distances, as in the case of MRK, the wake effects in TI at higher wind speeds are almost negligible.

Comparing the two wind speed bins, the wake effects in TI are much stronger in lower wind speeds as explained earlier.
Focusing on the years 2016-2019 we notice that the wake-induced TI from MRK is more sensitive to the wind speed compared to the one from BR1. This could be attributed to the smaller distance of MRK from FINO1, where the near wake region is more sensitive to changes of the thrust. Additionally, the larger size of the farm and the turbines in MRK could be a factor contributing to this observation.

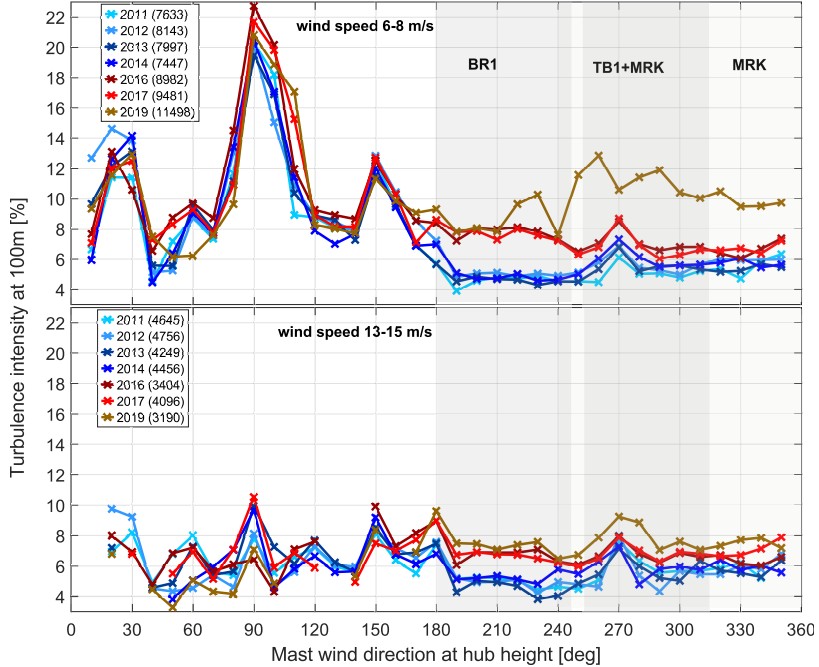

**Figure 7.** Turbulence intensity mean values per direction for the different years. The upper plot shows data for the 6-8 m/s wind speed bin and the lower for the 13-15 m/s bin. The shadowed areas indicate the azimuth sectors influenced by the denoted wind farms. The numbers in the legend indicate the amount of measurements used for each year

In order to examine how the wake induced TI is distributed, we looked at the sector-wise probability of occurrence of the
TI bins for a wind speed of 6-8 m/s (figure 8). In most cases, the probability distributions in the years 2016-2019 are shifted





towards higher TI levels while maintaining the distribution shape. This indicates that the overall levels of TI are increased and the results shown previously are not statistical artifacts from using the mean value.

For the sectors 215-235, we see an increase in 2019 which can be attributed to the small part at the edge of the MRK farm influencing FINO1. In the 235-245 sector, no increase occurred in 2019 to the mean value or the probability distribution again
due to the MRK layout. In sectors influenced by the MRK farm (sectors 245-325 degrees), the distributions are flatter with a less sharp peak. This is more apparent, for instance, in the directions of 250 and 300 degrees. The reason for that is not clear. A preliminary look at some of the 2020 data showed similar patterns. Thus, we believe that is not connected with some measurement issue. A correlation with the distance and size of the MRK farm could be possible but more research is required.

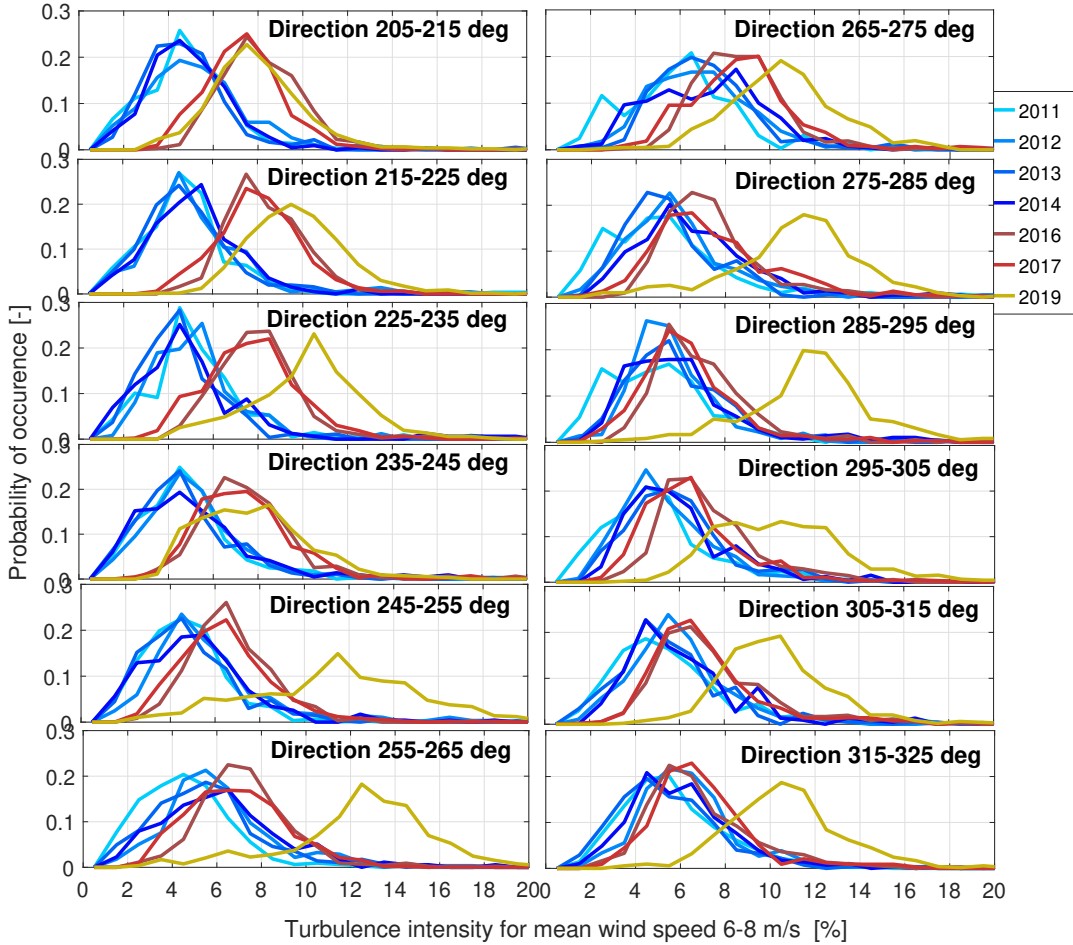

**Figure 8.** Turbulence intensity probability of occurrence for different directions and years for wind speeds 6-8 m/s.

The effect of the inter-farm interactions on the shear power law exponent ($\alpha$) was investigated. By examining the shear per
direction and wind speed we could not identify consistent patterns. We found a stronger correlation between $\alpha$ and TI. This can be seen in figure 9 where the fitted $\alpha$ is plotted against wind speed and TI for the azimuth sector of 200-320 degrees. The





mean shear exponent in the presence of farm wakes (years 2016-2019) is lower for all wind speeds up to 16 m/s. Looking at the band, we observe that the lower limit is constant over all periods while the upper is higher in the years 2011-2014 when AV was the only operating farm. This is directly caused by TI, which in the earlier period was lower. We verified this by filtering
with TI; when only values above 5% were kept the two curves and the bands matched (not shown here). The plot against TI, which includes all wind speeds, shows that the mean value and the band limits are very close for all periods for TI values higher than 4%. For lower TI levels, $\alpha$ is found lower for the later periods. Only a few measurements were available in this TI region, as events with so low turbulence are rare in the period 2015-2019 due to the wake effects and the fitting quality was not good as explained in the next paragraph. To conclude, the perceived wind shear as expressed by $\alpha$ was found to be decreased in the
presence of farm wakes due to the increased turbulence without clear correlations to distances and directions.

The use of the power law exponent to describe the wind shear is common practice in wind energy. Nevertheless, especially in wake situations, this is difficult as the shear is not only driven by the stratification of the atmosphere (i.e. by the temperature and pressure gradient) but also by the mechanical mixing introduced in the boundary layer due to the wakes. Then, the vertical wind profile cannot be adequately described by the power law making the exponent fitting procedure quite uncertain. This was
the case also in our study, where especially in directions affected by the wake of MRK which is very close to FINO1, the best fit did not match well the observed shape.

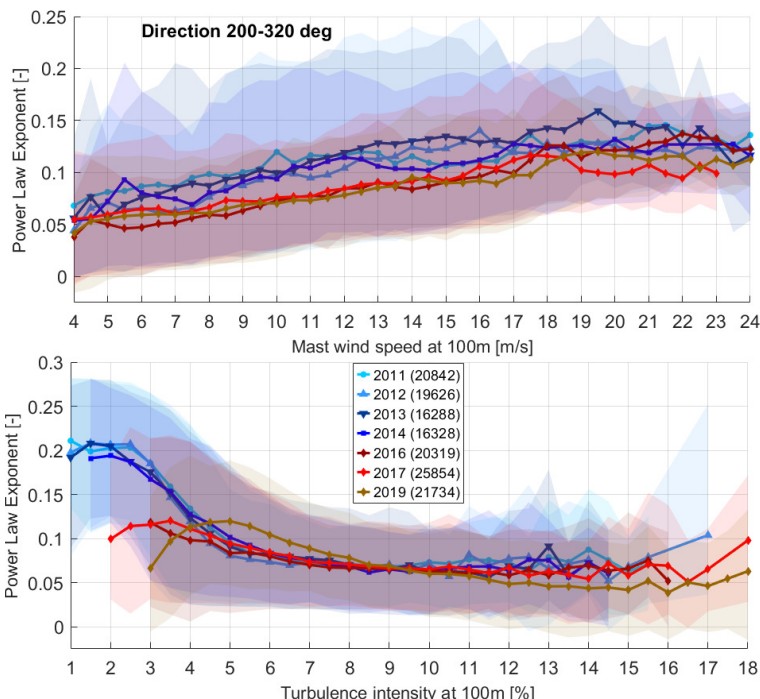

**Figure 9.** Fitted shear power law exponent at FINO1. Upper plot: Mean wind shear power law exponent versus wind speed, considering all levels of turbulence intensity. Lower plot: Mean wind shear power law exponent versus turbulence intensity including all wind speeds. The band represents the $15^{th}$ and $85^{th}$ percentiles. The numbers in the legend indicate the amount of measurements used for each year





To examine whether the local climate has changed over the years, we looked at the temperature and pressure measurements. The time series of temperature at different heights and pressure at 20 m a.s.l are shown in the appendix in figure A1. The trends are very similar for all the years indicating that the local climate has not changed. This suggests that the changes we observed in the wind conditions are attributed to inter-farm interactions. In figure 10 we show the probability of occurrence of the temperature difference between the water surface and the measurement at 50 m for the different years. This is used as an indicator of the temperature gradient distribution that drives atmospheric stratification. The distribution has not changed noticeably over the years. Only one year (2011) shows lower-than-average values.

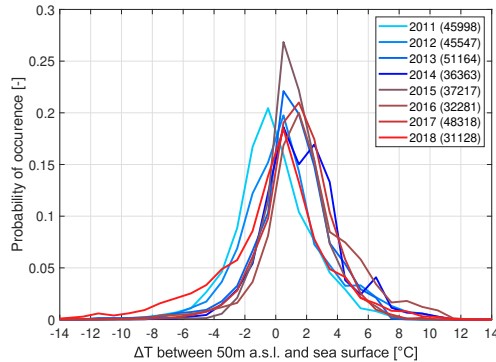

**Figure 10.** Probability of occurrence for temperature difference between 50 m a.s.l and sea surface for the different years. The numbers in the legend indicate the amount of measurements used for each year

The oceanic measurements of $H_s$ and $T_p$ were also analyzed to examine whether the conditions have changed over the years and whether the inter-farm wakes have any effect on these values. In figure 11 we show the site's wave characteristics in terms of $H_s$ per wind speed and $T_p$ per $H_s$ along with their linear fits. No correlation was found between the direction of wind and waves and $H_s$ and $T_p$ over the years. The measured time series are shown in the appendix in figure B1. Moreover, we couldn't identify any influence of the inter-farm interactions on these values in terms of magnitude, period or frequency of occurrence.

### 4.2 Turbine response

After observing the changes in the metocean conditions of AV, we look at how these affect the turbine's response. In figure 12 we present the tower bottom DEL in the fore-aft direction for the AV4 turbine for the different years and azimuth directions. As the data from 2019 are not included, the only relevant farms are TB1 and BR1. As discussed in section 4.1, the wakes of TB1 don't influence significantly the conditions at AV which is also seen in the loads. Hence, in the sector 260-320 degrees, the load response is not changing which also validates the consistency of the measurements.

In the rest of the sectors, the loads are increased in the below rated and the transition regions while at higher wind speeds the load level is similar (the rated speed of the machine is about 13 m/s). In the sector 200-220 degrees, we observe the larger differences at wind speeds 7-11 m/s with an increase of 5-40 %. At very low wind speeds, close to cut-in (about 4 m/s for this machine), the loads are driven by the controller's behavior, hence we don't see large deviations in the loads. At the transition

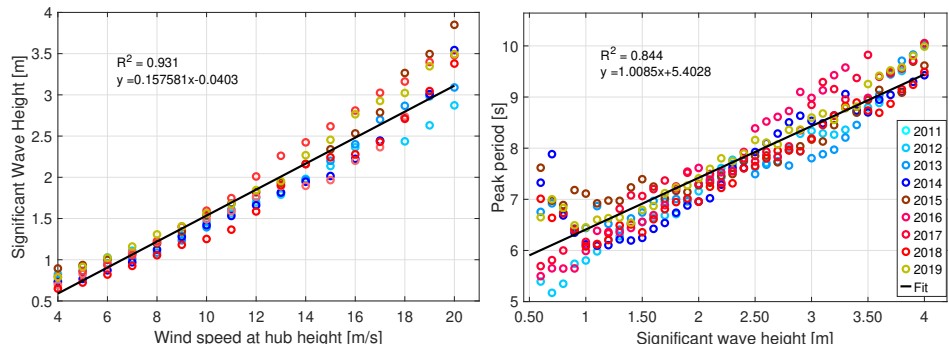

**Figure 11.** Left: Mean significant wave height per wind speed for the different years considering all directions. Right: Mean peak period over mean significant wave height for the different years. In both plots the linear fit is shown along with the relevant coefficient of determination. The numbers in the legend indicate the amount of measurements used for each year

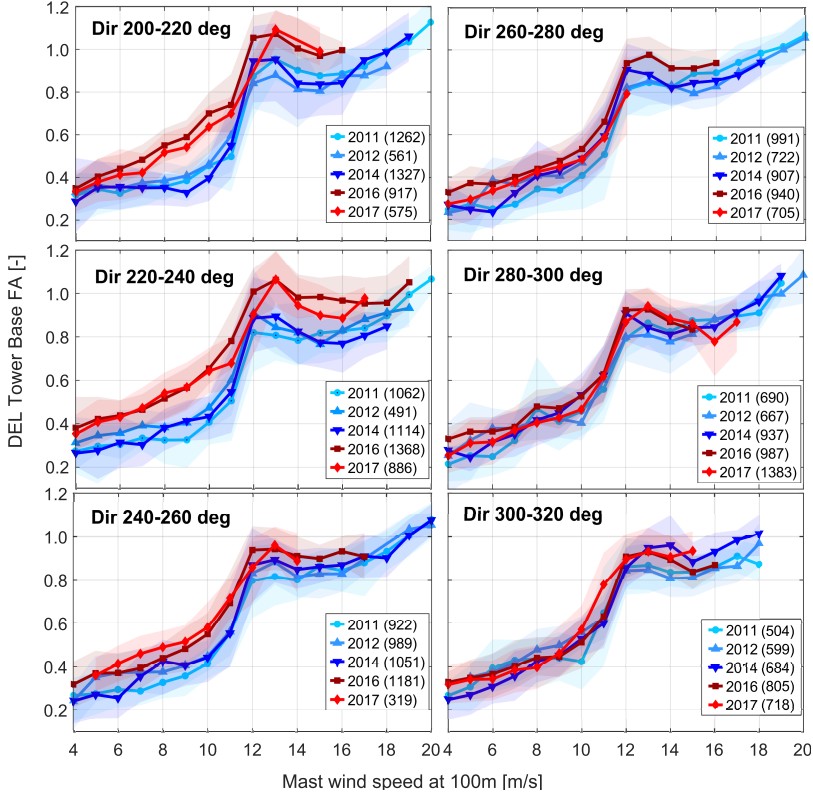

**Figure 12.** Mean value of damage equivalent loads at the tower bottom, in the fore-aft direction, against wind speeds for the different directions and years. The band represents the $15^{th}$ and $85^{th}$ percentiles. The numbers in the legend indicate the amount of measurements used for each year




region, in speeds 11-14 m/s, we observe load increase at a level of 10-20 %. This effect is reduced with the increase of the wind

speed and for wind speeds about 16 m/s and higher the load levels converge. Similar behavior is seen at the sector 220-240 degrees with similar load increases as AV is still in the full wake of BR1. In the sector 240-260 degrees, which is less influenced by BR1 we see a reduced effect in loads. The highest increases up to 30 % are found at wind speeds 8-11 m/s. At higher wind speeds, above 16 m/s, the loads are not affected.

In figure 13 we present the mean DEL per direction for a wind speed bin of 6-8 m/s. As discussed for the similar plot for the

TI (see figure 7), the sector of 10-190 degrees is influenced by the wakes of AV. Hence, this sector can only be used to evaluate the consistency of the load measurements over the years as the wind speed perceived by the turbine is expected to be higher than the measured value by FINO1. The influence of BR1 is seen at directions 200-250 degrees with the mean value, along with the percentiles, being shifted towards higher values for the years 2016 and 2017. Furthermore, the effect of the TB1 farm is minimal suggesting that at this distance and for this specific farm size and layout, the farm wakes don't affect the tower's

structural loading.

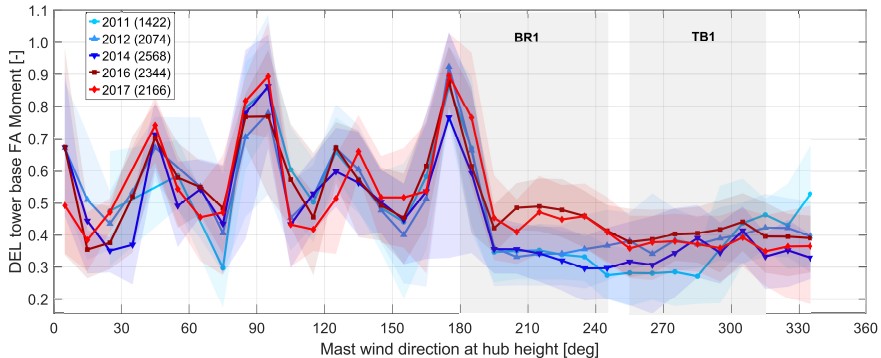

**Figure 13.** Mean value of damage equivalent loads at the tower bottom, for the fore-aft direction, per wind direction for the different years. Wind speed is 6-8 m/s. The band represents the $15^{th}$ and $85^{th}$ percentiles. The numbers in the legend indicate the amount of measurements used for each year. The shadowed areas indicate the azimuth sectors influenced by the denoted wind farms.

The SCADA data were also examined to identify possible correlations of the farm effects to the turbine's response. The power production was not found to be significantly affected by the increased TI. As a measure of the impact on the pitch system, we evaluated the standard deviation of the blade pitch angle from the SCADA data. In figure 14 we show these values per direction and wind speed. This signal is highly affected by minor changes to the controller's behavior and the condition of

the blades and pitch system. Nevertheless, a clear trend of increase is observed when the two periods (2011-2014 and 2015-2019) are compared. In the sectors 200-260 degrees, the effect of BR1 is seen with increase up to levels of 30%-40%. In the the sectors 260-300 degrees we see an increase in the pitch activity due to TB1. This shows that, although the tower loads don't seem to be affected by the small increase in TI in this sector, the pitch system is being more loaded.

We show the mean standard deviation of the generator speed signal over wind speeds in figure 15. Similar to the pitch signal,

this signal is also influenced by factors such as the controller settings and the condition of the drive train and generator systems

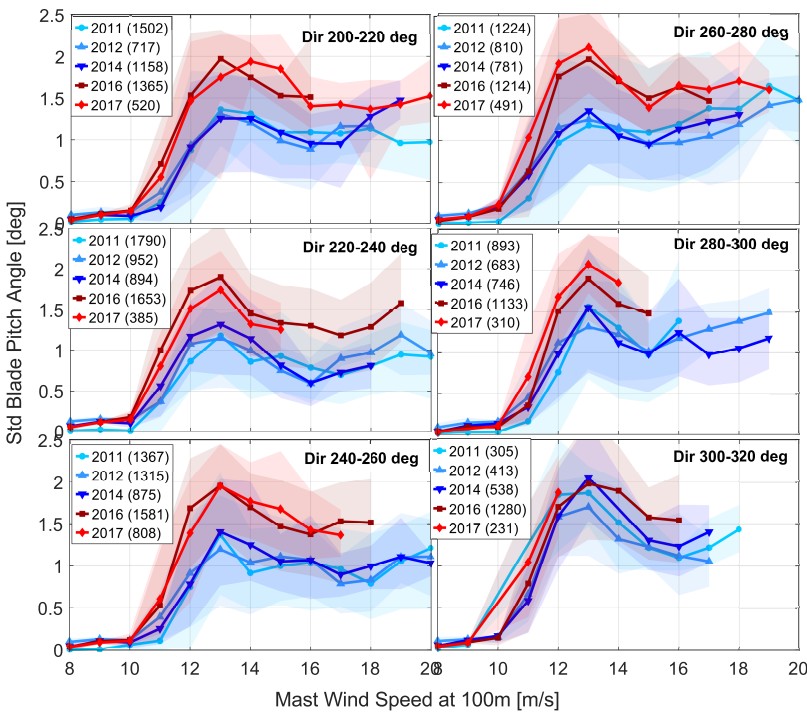

**Figure 14.** Mean standard deviation of the blade pitch angle per wind speed for different directions and years. The band represents the $15^{th}$ and $85^{th}$ percentiles. The numbers in the legend indicate the amount of measurements used for each year

leading to increased uncertainty and scatter of the measurements. Still, as in the previous case, we observe increased usage of the generator due to the increased TI levels. In sectors 200-260 degrees we see an increase of the generator speed variation up to 40% in below rated operation, as in higher wind speeds the regulation is handed to the pitch controller. To a smaller extent, although still visible, the effect of TB1 is seen in directions 260-300 degrees.

## 5   Discussion

Having a fully equipped offshore measuring platform, located near a farm and capturing data for long periods while the surroundings are changing, is rare and very beneficial for assesing offshore wind farm wake effects. Nevertheless, still a lot of uncertainties come into play as explained in section 3. The directional results are subject to uncertainty as the sectors of influence of the neighboring farms are not strict. This is due to possible errors in the exact locations of the turbines and the measurement equipment but more significantly influenced from the wake expansion itself and the possible meandering of the farm wake. Moreover, this could also be influenced by the turbine yaw misalignment, which is common to a level of some degrees, during normal operation. Additionally, as offshore farms are becoming larger reaching several kilometers and turbine

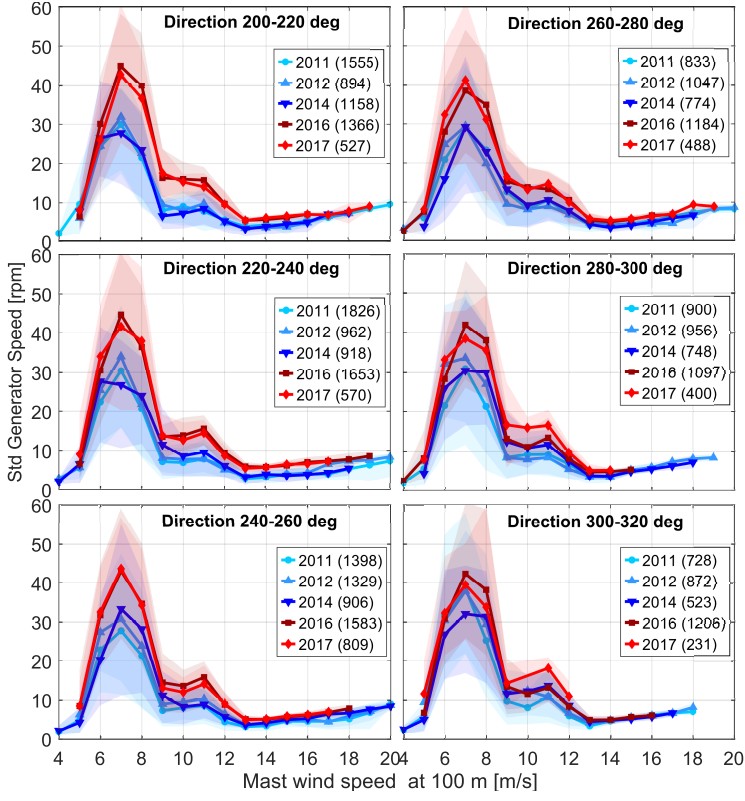

**Figure 15.** Mean standard deviation of the generator speed per wind speed for different directions and years. The band represents the $15^{th}$ and $85^{th}$ percentiles. The numbers in the legend indicate the amount of measurements used for each year

wakes are not linear features, a single point of measurement is probably not enough to fully characterize the inflow that could be varying across the wind farm.

Another important aspect relevant to this study is the consideration of atmospheric stability classification. It is well established by research that the wake effects (both at turbine and farm level) are directly correlated with stability. In this study, we decided not to add this dimension to the data processing as we focused on the analysis of the results cumulatively per year. The temperature difference results shown are an indicator that the annual stability distributions are similar over the different years. Moreover, the boundary layer is more affected by the mixing taking place through the interaction with the upstream

farms making it difficult to characterize atmospheric stability based on turbulence intensity or vertical shear. Common stability measures like Monin–Obukhov theory might not be applicable due to the wake (not ideal free-stream) conditions. The chosen approach allows us to examine the cumulative effect of the inter-farm interactions on the mean annual conditions experienced by the turbines. This is of practical interest when assessing or simulating the conditions and performance of a turbine at the site.





The findings here show the dependency of the perceived wake effects to the distance and size of the farms. We summarize here some of the points we observed that need more investigation. The operation of the farm BR2, located directly upwind of BR1 did not seem to affect further the measurements at FINO1 both in terms of wind speed and TI. Furthermore, the wake-induced turbulence from the closest farm MRK was found to be more sensitive to wind speeds. This could be correlated to the distance and also to the fact that the machines are larger than the other farms examined and operate with a larger variation on

the thrust coefficient. Additionally, the characterization and modelling of the vertical wind shear, especially when the distances between the farms are smaller, has to be evaluated. In general, more research is needed to understand the correlation between -wind farm size and layout, turbine capacity and size- and the strength of the wake effects, the distance they can propagate and their influence on the local micro-climate of the downstream farm.

    Another factor to consider is the weighting, in terms of probability of occurrence, of wind speeds and directions when

assessing the overall effect of the inter-farm interactions. As seen from the wind rose and the Weibull distributions, the wind speed and directional bins where the wake effects are stronger are the most probable ones. This means that the impact they have on lifetime loading and revenue is expected to be more significant when calculating the aggregated values. Considering this weighting is important in the cases where inter-farm wakes are accounted for in decision-making for processes like initial siting, maintenance planning, end-of-life strategies, operational and bidding strategies.

To facilitate further research on this topic, it could be very useful to have measurements from the inflow of both the upstream and downstream farms, ideally in a distributed manner. Moreover, at least some information such as operational status or power production from both interacting farms (upstream and downstream) would be very useful. Another aspect that could be worth investigating in future scenarios, with higher wind farm density and even closer spacing in favorable sites, is the coordinated operation of wind farm clusters. It may be that, by adjusting sector-wise the operation of the upstream farms -according to their

size, layout and relative orientation-, one could increase the overall power production and even reduce the overall structural loading.

## 6   Conclusions

In the present work we used metocean and turbine data to evaluate the effects of the inter-farm interactions at the offshore site Alpha Ventus. The nearby measurement platform FINO1 allowed us to examine how the local conditions have changed

from the period where AV was the only farm in the region to the later period where four larger wind farms started operating in distances varying from 1.4 km to 9 km. The data were analyzed on an annual basis and with regards to directions and wind speeds.

    Systematic changes in the flow conditions and the turbine response were found that can be attributed to wind farm wakes. This was validated by the agreement of trends and magnitudes in the values examined for the two distinct periods as well

the directional results. Moreover, the general atmospheric conditions were found not to be changed over the years in terms of temperature or pressure.





The wind speeds were found to be reduced, with the shape parameter of the fitted Weibull distributions being reduced by 15 % on average. For a theoretical 5 MW wind turbine this translates to 15-20 % reductions in AEP. The increase of measured turbulence intensity was found to be highly correlated with the distances between the farms, reaching a maximum mean

increase of 120 % for the closest farms (1.5 km) in below rated conditions. The vertical wind shear as expressed by the fitted power law exponent, was found to be reduced. Although, especially in the sectors related to the closest farms, it seemed to deviate from the power law approximation. The oceanographic measurements suggested no correlation between the presence of farm wakes and the values of significant wave height and wave period.

Turbine measurements were used to investigate how the changed flow conditions are perceived by the turbine of Alpha

Ventus closest to FINO1. The tower bottom fore-aft loads were influenced by the closer wind farm reaching an increased level of 30%-40% for below rated wind speeds. Both generator speed variation and blade pitch activity were found to be increased as well, even at sectors where loads did not change. This suggests that inter-farm wake effects have to be considered for all the systems and components of the turbine and not only for the structural loads.

The measurement results presented here show the possible technical and financial impacts of inter-farm wakes which have

to be considered when planning the siting of wind farm clusters. Moreover, it shows the need for analytical and simulation models that will be able to reproduce these effects taking into account the layout, sizing, and relative distances of neighboring farms.

*Code availability.* The in-house codes used for processing the data are not publicly available but can be requested by direct communication from the authors.

*Data availability.* Data cannot be shared publicly as they are covered by a user agreement with the RAVE consortium for Alpha Ventus. Data can be requested directly from the RAVE consortium and the BSH at https://www.bsh.de/EN/DATA/data_node.html.

## Appendix A

### A1  Temperature and pressure time series

In figure A1 we show the time series of water surface temperature, temperatures at 30 and 50 m a.s.l., and pressure at 20 m

a.s.l. for the different years. A moving average of 2 days is used for the results shown to make the seasonal trends clearer.

### A1  Oceanic data time series

In figure B1 we show the time series of the measured wave significant height and wave peak period for the different years. A moving average of 2 days is used for the results shown to make the seasonal trends clearer.



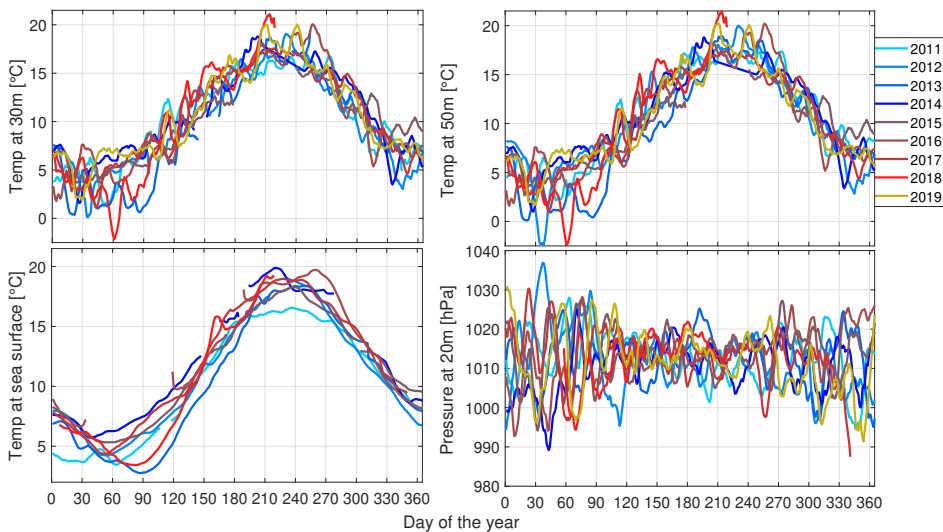

**Figure A1.** Temperature measurements at sea surface, 30m and 50 m and pressure measurements at 20 m over time for the different years

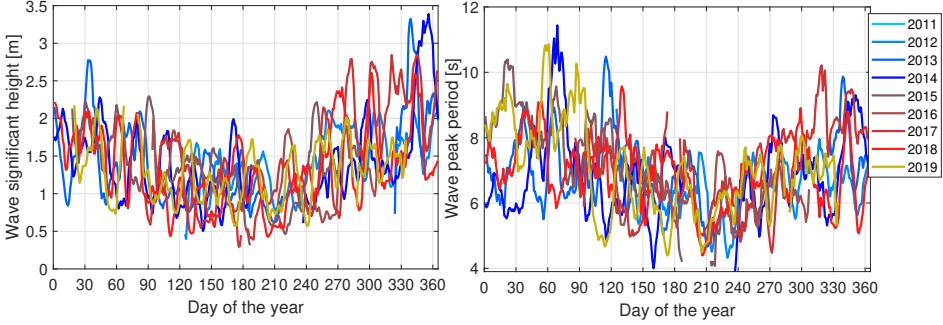

**Figure B1.** Peak period and significant wave height over time for the different years.

*Author contributions.* VP and MK conceived the idea of the research. VP performed the data processing, visualisation, and wrote the
manuscript. MK and VP developed the codes for data processing and visualization. VP, MK, AC and PWC revised the manuscript.

*Competing interests.* The authors declare no competing interests

*Acknowledgements.* This work is funded by the German Federal Ministry for Economic Affairs and Energy (BMWi) in the framework of
the national joint research project RAVE - OWP Control (ref. 0324131B) and is part of the research done in the WindForS research cluster.
We would also like to thank Senvion and DNV for providing the turbine measurements. Additional acknowledgements go to the Federal



Maritime and Hydrographic Agency (BSH) for providing access to the measurement database. Finally, we thank 4coffshore for allowing the use of their maps.





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
