# Peer review of "On the effects of inter-farm interactions at the offshore wind farm Alpha Ventus"

_Wind Energy Science, 2021_

## Community Comment (CC1)

Comments on the manuscript "On the effects of inter-farm interactions at the offshore wind farm Alpha Ventus" by V. Pettas et al.

This manuscript brings very interesting facts regarding the impact of newly build offshore wind farms on the alpha ventus wind farm in the German North Sea using long term data of the nearby met mast FINO1. Since offshore wind energy has been increasingly developed, this topic became very important.

The authors have shown clearly how the commissioned surrounding wind farms affect the wind conditions at FINO1 and, consequently, the loads of the turbine 04 of alpha ventus (AV04). They analyzed the annual changes of the atmospheric conditions measured at FINO1, and related them to the influence of the upstream wind farms. Finally, they concluded that the wake effects reduced the wind speed and increased the turbulence intensity at FINO1. Moreover, analyzing the loads on the tower base of AV04, they concluded that the damage equivalent loads (DEL) increased after the commissioning of the Borkum Riffgrund 1 wind farm (BR1), while the Trianel Windpark Borkum I wind farm (TWB I) didn't cause an increase in the fore-aft tower base DEL.

Having performed a similar analysis [1-3] we want to share our knowledge, experiences and results in this comment helping to further improve the manuscript.

For the next comments, the convention used includes information on the line number (*L. x) as printed in the mentioned manuscript.

L. 34

The review of relevant literature dealing with measurements of inter-farm wake turbulence and their load effect should be extended by previously published work, e.g. [2, 3].

L. 35

When discussing the work of Wu and Porté-Agel (2017) the classification of the atmospheric stability rather than the "vertical temperature gradients (K/km)" should be mentioned. See for instance Sorbjan and Grachev (2010) for guidance. Thermal stratification is driven by the gradient of the potential temperature rather the gradient of the absolute temperature. In offshore, the difference between sea surface and air temperature is of particular importance.

Sorbjan, Z. and Grachev, A. A.: An Evaluation of the Flux-Gradient Relationship in the Stable Boundary Layer, Bound.-Lay. Meteo- rol., 135, 385–405, https://doi.org/10.1007/s10546-010-9482-3, 2010.

L. 123

The strain gauges in alpha ventus were installed in half-bridge configuration and because of that the measurements are influenced by temperature. How have the authors managed the temperature influence on the strain gauges outputs for the calibration of the sensors, i.e. the calculation of the offset?

The authors have used 360° rotation of the nacelle to calibrate the strain gauges. The temperature on the tower base during this procedure can be very different from the mean temperature of the year (or the month), causing errors in the offset calculation. I would suggest a more detailed explanation of how the authors have worked with this uncertainty.

We have done a very similar analysis for turbine AV07 [1]. We couldn't define an offset for the strain gauges when considering full 360° rotations of the nacelle because the variation between the

calibration trials was too high. We found a more consistent value when selecting many periods when the turbine was off and stationary.

L. 127

We suppose that the availability of the turbine's data is lower than the one of FINO1's data. In order to avoid a bias in your results, the selected data should represent the normal atmospheric conditions, i.e. the ones shown by all available data from FINO1. One example is the relation between wind speed and turbulence intensity (TI). After applying the filtering on the data of AV04 and dividing them into sectors, the TI per wind speed was probably not the same as the one obtained from FINO1 data. How have the authors guaranteed that the selected data were representative? We would suggest a further filter to make the atmospheric conditions of the selected data more similar to the normal ones.

L. 204

Annual meteorological trends, which can cause an increase or reduction of the yearly-mean wind speed, make it more complicated to analyze the wake effects on the wind speed in each year. It would be interesting to show if the annual variations of wind speed are not caused by annual meteorological trends. That could be done by analyzing long term data, e.g. from the New European Wind Atlas [4], as it was done in [1]. Averaging the years into periods would also reduce the effect of annual meteorological trends in the wake effect analysis.

L. 213

In the present analysis the wind farm Trianel WindPark Borkum I seems to not influence the wind speed at FINO1, however another analysis showed that the wind speed in the sector of this wind farm (255° - 310°) was indeed reduced after its commissioning [1-2].

The results were achieved by grouping the years into phases according to the commissioning date of the wind farms surrounding FINO1, as shown in Figure 1.

[Figure]

*Figure 1 - Timeline of the measurement phases [3].*

Figure 2 shows the reduction in wind speed in the sector 225° - 310° for phase 3, i.e. half end of 2015 to half start of 2019 (cf. Fig. 1).  Could the author explain why a similar behavior is not seen in Figure 5 of their manuscript?

[Figure]

*Figure 2 - Polar plot with wind speed as parameter for measurement phases 1, 2 and 3, with the projection of FINO1 and the wind farms alpha ventus, Borkum Riffgrund 1 and Trianel Windpark Borkum I [1].*

L. 292

The authors claimed that the wind farm Trianel Windpark Borkum I doesn't influence significantly the conditions at alpha ventus and its loads. Since an increase in turbulence intensity at FINO1 was caused by the wakes of TWB I, it is expected that they would also cause an increase in the 10-minute damage equivalent load per wind speed. One of the reasons that could explain the persistent DEL in AV04 after the commissioning of TWB I is the lack of representativeness of the selected data, as it was pointed out for line 127.

Another option could be the change in the control strategy of alpha ventus that was implemented in 2015. The turbine AV07 had its rated power reduced from 5.3 MW to 5.1 MW [3]. In the analysis with AV07 data, this change in the rated power seemed to affect only the wind speeds above rated wind speed, causing a reduction in the DEL of the tower base fore-aft bending moment, although the TI increased, see Figure 3.

The section used for Figure 3 was 203° - 317°, i.e. it also includes the wake effects from Borkum Riffgrund 1 wind farm.

[Figure]

*Figure 3* - Bin-mean of damage equivalent load of the normal bending moment on the tower base as a function of wind speed during phases 2 and 3 (cf. Fig. 1) [3].

Have the authors checked if the turbine AV04 experienced any change in the power controls (like it was applied to AV07) that could influence the loads?

Technical comment

We would suggest a general revision of the correct names of the wind farms, for example "Trianel Windpark Borkum I" instead of "Trianel Borkum 1".

We hope that these comments can help to further improve the manuscript and that they are considered in a revised version.

Best regards

Marcos Ortensi

**Bibliography**

[1]     Ortensi, M., "Wind farm wake effects on the wind conditions and the fatigue loads of the offshore wind farm Alpha Ventus", master thesis, University of Oldenburg, 2020.

[2]     Ortensi, M., Fruehmann, R, Neumann, T., "Long-term Effects of Wakes from Offshore Wind Farms on Wind Conditions at FINO1", *white paper UL International,* November 2020. https://aws-dewi.ul.com/knowledge-center/item/long-term-effects-of-wakes-from-offshore-wind-farms-on-wind-conditions-at-fino1/

[3]     Ortensi, M., "Wind farm wake effects on the wind conditions and the fatigue loads of the offshore wind farm Alpha Ventus", *RAVE Workshop*, 2021. https://rave-offshore.de/files/downloads/konferenz/Workshop-2021/Ses2_2_RAVE2021_alphaVentus_Ortensi.pdf

[4]     "New European Wind Atlas," New European Wind Atlas NEWA, [Online]. Available: https://map.neweuropeanwindatlas.eu/.

---

## Author Comment (AC1)

**Authors' Response (manuscript wes-2021-50)**

We thank all reviewers and commenters for their constructive comments that helped improve the quality of the manuscript and appreciate the time and effort they put into this. In the following pages, we reply to the comments on a point-by-point basis. An annotated version of the revised manuscript, including all changes suggested by the commenters as well as editorial changes by the authors, is attached at the end of the present document.

On behalf of the authors,

Vasilis Pettas

**Reply to RC-1**

The authors would like to thank the reviewer, Nicolai Gayle Nygaard, for his time and useful feedback. All the comments have been taken into consideration and have contributed to improving the manuscript. A list of point-by-point replies to the comments follows (reviewer comments in black, authors' response in blue).

**General comment:**

As a general comment, I do not understand the inclusion of fitted shear exponents based on power law wind speed profile. In a waked flow the power law cannot be expected to apply, as the wind speed deficit in the wake modifies the wind speed profile. There are many examples of this in the literature. The authors even indicate this in line 274. Reporting a shear coefficient is meaningless if it results from a poor fit to power law profile. I therefore suggest that all discussions on shear are removed from the manuscript. Alternatively, the authors should add quantitative details on the quality of the fits along with the reported shear coefficients in Figure 9 and elsewhere.

We agree with the reviewer that the characterization of vertical shear profile with a power law is not optimal for wake conditions as we have stated in the manuscript. Nevertheless, we believe that the information should stay in the manuscript as it gives an impression of the overall conditions on the site for the different periods. Moreover, we believe that it can be useful input for practiotioners wanting to do aeroelastic simulations using these data as commonly the power law is used in these cases to describe the vertical shear. To our knowledge, there is no general method to characterize the vertical wind profile using height measurements in wake situations. We believe that this is a topic worthy of further investigation but is not in the scope of the current study. We have added the mean and std of coefficient of determination for the PL fitting and discuss this more in the relevant paragraph, according to the recommendations from the reviewer.

**Specific comments:**

Include a reference to Ortensi, Frühmann and Neumann, Long-term Effects of Wakes from Offshore Wind Farms on Wind Conditions at FINO1, UL white paper, 2020

**The reference has been added and discussed in the literature review.**

L9: wind turbine performance typically relates to the power production. Since the paper is investigating the loads and how they affected by the inter-farm effects using the term turbine performance is somewhat misleading.

By the term performance, we referred to the generator and pitch metrics included in the study. We have changed the wording from turbine response to "generator and pitch activity" to make it more specific.

**L100: please include examples of the corrections applied to the FINO1 data (or a reference). This will increase the reproducibility of the analysis**

As the FINO1 database has been dynamically changing over the years with new correction and calibration values these changes have been added gradually to our internal database. The University of Stuttgart is part of the RAVE consortium since the beginning of the project and we have been applying these changes over time. Most of them are now incorporated directly into the RAVE database. The effects include mostly shadowing effects affecting wind speed measurements and wind direction results. We updated the text to give an example and also refer to some of the early publications explaining these effects.

**L123: please explain in further detail how the calibration of the nacelle yaw sensor was done.**

For the nacelle yaw position sensor, a slow drift over time was observed. This was corrected by using events where the turbine was not operating and the nacelle was rotated 360 degrees; by correlating the known locations of the strain gauges around the tower and the nacelle yaw signal corrections of the offset were derived and the drift compensated. We updated the text in the manuscript to explain this.

**L125: have the small corrections on the calibration factors been described elsewhere? Otherwise, please include further details**

Please refer to our clarification in the previous comment. We have submitted our corrections to the measurement data provider (DNV) so that the values can be included directly in the RAVE measurement database and future research can profit from it.

**L128: specify the appropriate thresholds, e.g. in an appendix**

We included the thresholds in the main text stating the exact values for wind speed, TI, and nacelle yaw. Regarding the turbine response (generator speed, power, loads, pitch) we did not include the values as they cannot be disclosed according to the agreement with the manufacturer.

**L130: is there a directional pattern in the difference between the met mast wind direction and the turbine yaw direction? See examples of this in Schepers et al, Wind Energy 2012; 15:575–591**

We could not identify a specific pattern on the difference between met mast direction and turbine sensor. The reason we did this filtering was to avoid loads been biased by the yaw misalignment of the nacelle. As the focus of the paper is on the effects of inter-farm wakes on loads we didn't try to go deeper into the potential reasons for this misalignment.

L178: I agree that the wake from AV itself is expected to lead to an underestimation in the FINO1 wind speeds. But why not confirm that by comparing wind speeds in the affected sector in the periods before

**and after the construction of AV? The FINO1 mast has several years of measurements before the construction of AV.**

As the focus of the paper is on the inter-farm wake effects we decided to focus on periods where AV is operating (alone or with surrounding farms). We used the FINO1 data from the sector influenced by AV only to show that the (although affected by AVs own wakes) measurements are consistent through the periods and not to investigate the shadowing effects of AV to FINO1. As the paper is already quite loaded with figures and information we decided not to include preconstruction data from FINO1 or a discussion on AVs shadow effects to the mast to keep the focus.

**L183: please be more specific when talking about significant data gaps. How large are they?**

We made sure each sensor analyzed on an annual basis has at least 80% (after filtering) availability in order to be accepted. On top of this criterion, we made sure that the missing gaps are not concentrated in one period (e.g. a year missing in total 60 days of data spread out in smaller periods over the year would be accepted but a year missing two consecutive months (e.g. Nov-Dec) would be rejected) to avoid seasonal bias on the results. In the case of the year 2018, we had no wind data for the last 4 months of the year (in total we had about 60% availability). In 2015 we had less data missing (15% in total) but in combination with the explanation given in the next sentence of the manuscript (BR1 and TB1 started operating during the year in dates we were not able to verify or deduct from the data), we decided to exclude this year from the annual results. We updated the text to make this more clear and added the numbers.

L190: constant offset – this strictly speaking only applies if the wind speed deficit from AV is constant and the frequency of wind directions in the sector affected by AV is the same in all years

We agree with the reviewer, we changed the phrasing to: "This is not expected to influence the relative differences over the years since it has a low probability of occurrence and can be seen as an offset influenced only by the inter-annual variability."

L193: the AEP is the convolution of the power curve with the Weibull distribution (or more generally the wind speed distribution), not the product. Typically, the convolution is replaced by a discrete approximation summing the product of the two curves over all wind speed bins

We updated the text to explain the process more clearly as suggested by the reviewer. "The theoretical AEP is simply derived by summing the product of the discreet approximations of the theoretical power curve of the NREL 5MW reference wind turbine with the fitted Weibull distribution over all wind speed bins."

**L230: please include a reference to theory supporting the statement relating TI in the wake to the thrust coefficient of the upstream turbine**

We added the following two references to back up the statements in this sentence:

- 1. Frandsen, S. T.: Turbulence and turbulence- generated structural loading in wind turbine clusters, Ph.D. thesis, Risoe National Laboratory, 2007
- 2. Quarton, D. C. and Ainslie, J. F.: Turbulence in Wind Turbine Wakes, Wind Engineering, 14, 15–23, 1990.

**L236: what is a blizzard cage structure? Is it on FINO1?**

It is a metal structure for the weather protection equipment at the top of FINO1. This is explained also in Westerhellweg 2010. We repeated the citation here to assist the reader to look for more information on the topic.

**L247: why would the larger size of the wind farm MRK and its turbines be a factor? Make the argument clear to the reader**

We added further explanations to the argument to make it clearer for the reader. We also added a comment about further research required to identify the underlying mechanism for the higher sensitivity of wake-induced turbulence to wind speed for the different distances and farm sizes observed in the measurements.

**Figure 7: why is there no shading indicating the sector affected by wakes from MRK?**

As the overlapping shades were difficult to distinguish, we changed the way we indicate the sectors in both figures (7 and 13) to arrows and dashed lines to make the distinction clearer.

L305: the sector influenced by wakes from AV is described as 30-170 degrees on pages 8 and 11, why is it different here?

**We corrected the typo.**

L320: consider a better word than usage of the generator. I think you mean increased wear due to increased fluctuations of the generator speed

**We updated the text to clarify according to the suggestion.**

L331: the authors state that yaw misalignment is common to a level of some degrees. Please be more quantitative: how common? How many degrees? Add references to support this statement

This is an observation we have by working with different projects with operators and OEMs. In our experience when precise measurements are done correlating the mean wind direction upstream of a wind turbine with the yaw sensor there are is often an offset of some degrees (often in the range of 2-10 degrees). Of course, this varies depending on factors like location, age of hardware and wind turbines, etc. As we don't have studies to back this statement quantitively, we changed the phrasing in the manuscript to "Moreover, this could also be influenced by a possible yaw misalignment of the upstream turbines."

L341: Monin-Obukhov theory is not a stability measure. Do you mean the Monin-Obukhov length? Nonetheless, I agree that similarity theory likely does not apply in the wake.

We corrected the phrasing according to the suggestion. As the reviewer suggested the point of this sentence is to show that the effects we see are not stratification driven but wake driven.

L356: the conclusion of coincidence between the direction of strongest inter-farm wakes and the predominant wind directions is particular to this site. It is not a general conclusion

We agree with the comment. We updated the relevant text to clarify this.

L358: the weighting/accounting for inter-farm wakes is done routinely in the industrial application of wake models.

We agree with the reviewer the sentence aims to highlight the significance of this procedure. Since we don't perform this weighting in the paper we mention this to make the reader aware that the impact can be even larger than what is reported in the unweighted measurements. We changed the phrasing including also the previous comment to avoid misinterpretations.

Consider adding references to other papers analysing wakes between offshore wind farms, for example Hansen et al, Journal of Physics: Conference Series 625 (2015) 012008 and Schneemann et al Wind Energy Science 5, 29–49, 2020.

The suggested references were added and discussed in the literature review part of the paper.

**Reply to RC-2**

We would like to thank the reviewer for the positive recommendation as well as the time and effort to review the manuscript.

**Reply to CC-1**

The authors would like to thank the commenter, Marcos Ortensi, for the time and effort he put into suggesting improvements to the manuscript. A list of point-by-point replies to the comments follows (comments in black, authors' response in blue).

L34 The review of relevant literature dealing with measurements of inter-farm wake turbulence and their load effect should be extended by previously published work, e.g. [2, 3].

Thanks for pointing out this literature. We have included and discussed the relevant reference in the literature review part of the paper.

L35 When discussing the work of Wu and Porté-Agel (2017) the classification of the atmospheric stability rather than the "vertical temperature gradients (K/km)" should be mentioned. See for instance Sorbjan and Grachev (2010) for guidance. Thermal stratification is driven by the gradient of the potential temperature rather the gradient of the absolute temperature. In offshore, the difference between sea surface and air temperature is of particular importance.

Sorbjan, Z. and Grachev, A. A.: An Evaluation of the Flux-Gradient Relationship in the Stable BoundaryLayer, Bound.-Lay. Meteo- rol., 135, 385–405, https://doi.org/10.1007/s10546-010-9482-3, 2010.

We agree on the definition of stratification, in the cited paper they used the gradients to characterize the different conditions. Hence, we used the term vertical temperature gradients (K/km) as it is used in the paper itself to describe the different cases examined so that the reader can be easier guided if looking through the reference.

L123 The strain gauges in alpha ventus were installed in half-bridge configuration and because of that the measurements are influenced by temperature. How have the authors managed the temperature influence on the strain gauges outputs for the calibration of the sensors, i.e. the calculation of the offset?

The authors have used 360° rotation of the nacelle to calibrate the strain gauges. The temperature on the tower base during this procedure can be very different from the mean temperature of the year (or the month), causing errors in the offset calculation. I would suggest a more detailed explanation of how the authors have worked with this uncertainty. We have done a very similar analysis for turbine AV07 [1]. We couldn't define an offset for the strain gauges when considering full 360° rotations of the nacelle because the variation between the calibration trials was too high. We found a more consistent value when selecting many periods when the turbine was off and stationary.

We are aware of the issue on AV7 (ADWEN machine) regarding the thermal compensation on the strain gauges measurements. In the case of AV4 (Senvion machine), thermal compensation is included in the strain gauges. This is verified by direct communication with the data providers from DNV but also looking at the data for fluctuations related to the temperature. The calibration of the data has also been verified for inconsistencies and used in other works (see e.g. [1]). Finally, the consistency of the load measurements can be verified by looking at the sectors unaffected by the farm wakes (10-170 degrees). More information for the AV7 measurements that might be useful can be found in reference [2].

L127 We suppose that the availability of the turbine's data is lower than the one of FINO1's data. In order to avoid a bias in your results, the selected data should represent the normal atmospheric conditions, i.e. the ones shown by all available data from FINO1. One example is the relation between wind speed and turbulence intensity (TI). After applying the filtering on the data of AV04 and dividing them into sectors, the TI per wind speed was probably not the same as the one obtained from FINO1 data. How have the authors guaranteed that the selected data were representative? We would suggest a further filter to make the atmospheric conditions of the selected data more similar to the normal ones.

As we mention in the manuscript, the availability of the different measurements varies depending on the combination of sensor(s) availability and the filtering approach. In the manuscript, we correlate wind speeds to turbine responses based on the cumulative filtering. E.g. the wind speed from FINO1 and the yaw direction, controller status, power output, load sensor(s) from AV4 are combined to determine the wind speed binning and loads shown. The fluctuation of availability of all sensors is expected to vary randomly across all years, periods, the combination of sensors, etc. and not to create a systematic bias on the results. Moreover, as it is shown from the validation sectors the values are consistent in terms of mean values and percentiles over the years, with only small fluctuations seen attributed to the inter-annual variability.

L204 Annual meteorological trends, which can cause an increase or reduction of the yearly-mean wind speed, make it more complicated to analyze the wake effects on the wind speed in each year. It would be interesting to show if the annual variations of wind speed are not caused by annual meteorological trends. That could be done by analyzing long term data, e.g. from the New European Wind Atlas [4], as it was

done in [1]. Averaging the years into periods would also reduce the effect of annual meteorological trends in the wake effect analysis.

We cover the topic of inter-annual variability by analyzing the measurements on an annual basis and showing the differences between the years. Averaging multiple years and all speeds can blur the picture due to seasonal bias, availability, and other factors. We believe for the scope of our study classifying the years in data and binning all the quantities fits the purpose better. The use of the validation sectors (see e.g. figures 7 and 13) shows that the changes we see are not related to interannual variability but correlated to the presence of the upstream farms. Finally, other studies (e.g. see [3]) have shown a long-term inter-annual variability lower than 4.5% in wind speed magnitude for the north sea which is also our observation by comparing the years in the different periods (2011-2015 and 2016-2019). We have added the reference in the text and extended the discussion in the manuscript.

L213 In the present analysis the wind farm Trianel WindPark Borkum I seems to not influence the wind speed at FINO1, however another analysis showed that the wind speed in the sector of this wind farm (255° - 310°) was indeed reduced after its commissioning [1-2]. The results were achieved by grouping the years into phases according to the commissioning date of the wind farms surrounding FINO1, as shown in Figure 1. Could the author explain why a similar behavior is not seen in Figure 5 of their manuscript?

The results in the plot posted here are an average of all wind speeds and all years. We don't have more information on the procedure followed to comment further, as the sources are not public. Our approach was to systematically investigate the data on an annual basis and with wind speed bins. We believe that it gives a good insight into the inter-farm wake effects which is the main topic of this work. To summarize we think that the discrepancy between figure 5 in the manuscript and the plot posted here is due to different approaches and they should not be directly compared as they have different assumptions, use different filtering approaches and show different quantities (fitted Weibull distributions per year and direction vs mean speed of all wind speeds and multiple years).

L292 The authors claimed that the wind farm Trianel Windpark Borkum I doesn't influence significantly the conditions at alpha ventus and its loads. Since an increase in turbulence intensity at FINO1 was caused by the wakes of TWB I, it is expected that they would also cause an increase in the 10-minute damage equivalent load per wind speed. One of the reasons that could explain the persistent DEL in AV04 after the commissioning of TWB I is the lack of representativeness of the selected data, as it was pointed out for line 127.

Another option could be the change in the control strategy of alpha ventus that was implemented in 2015. The turbine AV07 had its rated power reduced from 5.3 MW to 5.1 MW [3]. In the analysis with AV07 data, this change in the rated power seemed to affect only the wind speeds above rated wind speed, causing a reduction in the DEL of the tower base fore-aft bending moment, although the TI increased, see Figure 3.

The section used for Figure 3 was 203° - 317°, i.e. it also includes the wake effects from Borkum Riffgrund 1 wind farm.

As mentioned in the manuscript we don't see any changes in the sectors not influenced by the neighboring farms. We also believe that the data are representative, as explained in a previous comment. We don't see significant changes in the controller behavior of the AV4 turbine affecting the loads. This is supported

by the validation sectors showing no significant changes in the loads and the power curves we obtained from the data. Moreover, we have worked extensively with these data and turbine models and have only seen control changes for preventive maintenance purposes that had, as a result, to filter out the year 2013 from the study due to an extensive period of curtailed operation (see e.g. [1] and [4]). Finally, we believe that the directional analysis in 20-degree sectors is better fitted for analyzing inter farm wake effects than large sector averaging, as the neighboring farms have different sizes/turbines, and distances from AV and the results can get more blurred when the sectors are examined cumulatively.

**Technical comment**

We would suggest a general revision of the correct names of the wind farms, for example "Trianel Windpark Borkum I" instead of "Trianel Borkum 1".

The purpose of using the names of the surrounding farms is to orientate the user around the map and the relative directions. To avoid long repetitive mentions, we have used the farm names excluding the part Wind Farm or wind park from the name as we believe it is clear enough to the reader.

[revised manuscript text omitted]